# A Robust and Sensitive Spectrophotometric Assay for the Enzymatic Activity of Bacterial Adenylate Cyclase Toxins

**DOI:** 10.3390/toxins14100691

**Published:** 2022-10-08

**Authors:** Marilyne Davi, Mirko Sadi, Irene Pitard, Alexandre Chenal, Daniel Ladant

**Affiliations:** 1Biochemistry of Macromolecular Interactions Unit, Department of Structural Biology and Chemistry, Institut Pasteur, Université Paris Cité, CNRS UMR 3528, 75015 Paris, France; 2Université Paris Cité, 75014 Paris, France; 3Structural Bioinformatic Unit, Department of Structural Biology and Chemistry, Institut Pasteur, Université Paris Cité, CNRS UMR 3528, 75015 Paris, France; 4Université Paris Sorbonne, 75231 Paris, France

**Keywords:** adenylate cyclase toxin, *Bordetella pertussis*, cyclic nucleotide, cAMP, spectrophotometric enzymatic assay

## Abstract

Various bacterial pathogens are producing toxins that target the cyclic Nucleotide Monophosphate (cNMPs) signaling pathways in order to facilitate host colonization. Among them, several are exhibiting potent nucleotidyl cyclase activities that are activated by eukaryotic factors, such as the adenylate cyclase (AC) toxin, CyaA, from *Bordetella pertussis* or the edema factor, EF, from *Bacillus anthracis*. The characterization of these toxins frequently requires accurate measurements of their enzymatic activity in vitro, in particular for deciphering their structure-to-function relationships by protein engineering and site-directed mutagenesis. Here we describe a simple and robust in vitro assay for AC activity based on the spectrophotometric detection of cyclic AMP (cAMP) after chromatographic separation on aluminum oxide. This assay can accurately detect down to fmol amounts of *B. pertussis* CyaA and can even be used in complex media, such as cell extracts. The relative advantages and disadvantages of this assay in comparison with other currently available methods are briefly discussed.

## 1. Introduction

Cyclic Nucleotides Monophosphates (cNMPs) are key messengers in most cell types that have been implicated in the modulation of numerous physiological processes [1]. Many pathogens have evolved sophisticated strategies to disrupt cNMP metabolism in the hosts they attempt to colonize [2]. In particular, several bacterial pathogens produce toxins that are endowed with endogenous nucleotidyl cyclase activity [3]. These toxins are able to invade eukaryotic cells where they are stimulated by endogenous co-factors to produce large amounts of cNMP, thus disrupting cell signaling and altering cell physiology. To date, the best characterized types are the adenylate cyclase (AC) toxin, CyaA, from *Bordetella pertussis* [4,5] and the edema factor, EF, from *Bacillus anthracis* [6]. They are key virulence factors that target the immune cells to promote efficient bacterial colonization of the hosts [7,8,9]. Both CyaA and EF are potently activated by the eukaryotic protein, calmodulin (CaM), and have very high catalytic efficiency (with kcat > 1000 s^−1^). Their catalytic moieties share remarkable structural similarity, while these two toxins differ markedly in their sequences, their secretion mechanisms, and their modes of invasion of eukaryotic target cells [3]. A distinct sub-family of virulent factors with nucleotidyl cyclase activity [10,11] has been recently characterized as being activated by eukaryotic actin [12]. It includes the ExoY toxin that is a type-3 virulent effector produced by *P. aeruginosa* and various ExoY-like modules present in different MARTX (Multifunctional-Autoprocessing Repeats-in-Toxin) toxins produced by certain *Vibrio* species as well as by other Gram-negative bacteria [3,13].

The characterization of these virulent factors frequently requires the determination of their enzymatic activities, which is the conversion of adenosine triphosphate (ATP) into cyclic AMP (cAMP) and pyrophosphate (PPi). Indeed, numerous studies have been devoted to the identification of critical residues in the catalytic reaction and/or the binding of the toxins to their eukaryotic co-factors [2,3]. Simple and rapid assays for determining enzymatic activities of recombinant enzymes harboring specific mutations would greatly facilitate such investigations.

One of the most widely applied in vitro techniques uses a radioactive substrate [α-^32^P or α-^33^P]-ATP that is converted into a radioactive cAMP product that is subsequently isolated by chromatography on neutral alumina and quantified with a scintillation counter. A main drawback of this approach is that it requires specific equipment and laboratory premises to safely handle radioactivity. To avoid this major hurdle, a variety of alternative enzymatic assays have been set up, including the measurement of cAMP with ELISA, use of fluorescent ATP substrates, measurement of the pyrophosphate with colorimetric of fluorimetric coupled assays, or measurement of the reverse reaction (eg cAMP + PPi -> ATP) with colorimetric assays (see Table 1).

Here we describe a simple and robust in vitro assay for AC activity based on the spectrophotometric detection of cAMP after chromatographic separation on aluminum oxide. This assay is cost-effective and straightforward for routine monitoring of AC toxin activity and sensitive enough to detect down to fmol amounts of *B. pertussis* AC. Moreover, with proper controls, it can also be applied to quantify AC in complex mixtures, such as cell extracts. The relative advantages and disadvantages of this assay in comparison with other methods (Table 1) are presented in the Discussion section.

## 2. Results

### 2.1. Spectrophotometric Detection of AC Activity

Chromatography over aluminum oxide has been widely used for the separation of cyclic nucleotides from other nucleotides (NTP, NDP, or NMP), as these latter nucleotides are selectively adsorbed on Al_2_O_3_ [26,27]. In the following assay procedure, ATP is converted into cAMP by AC in a first step, and in a second step, cAMP is separated from ATP by chromatography on Al_2_O_3_; at pH 7.2–7.6, deprotonated ATP (ATP^4−^, as well as ADP^3−^ and AMP^2−^) is retained on the alumina, whereas cAMP^−^ is not (provided the ionic strength of the elution buffer is higher than 0.1 M NaCl—data not shown). Thus, the alumina eluent exclusively contains cAMP, which can be easily determined by measuring the absorption at 260 nm, given the high absorption coefficient of cAMP (molar extinction coefficient, ε_260nm_ = 15,000 M^−1^·cm^−1^). When using relatively pure AC preparations or highly-active ACs like bacterial AC toxins, there are no other contaminating molecules that absorb light at 260 nm, so the assay is very straightforward and robust. In standard conditions, the reaction is carried out at a 0.1 mL final volume with 2 mM ATP for appropriate time (from 5 min up to several hrs). Incubation is arrested by directly adding 0.3 g of Al_2_O_3_ dry powder to the reaction mixture (alternatively, an excess of EDTA can be added to chelate Mg^2+^, an essential co-factor, before transferring the reaction mixture into pre-weighted Al_2_O_3_ dry powder). After the addition of 0.9 mL of HBS buffer (50 mM Hepes, pH 7.5, 0.2 M NaCl), the mixture is incubated for a few minutes with agitation, the tube is centrifuged (5 min at 10–13,000× *g*), and the absorption at 260 nm (A_260_) of supernatant is recorded. When multiple assays are carried out in parallel, it is convenient to transfer supernatant (e.g., 0.3 mL) into a UV-transparent microplate to record the absorption with a microplate reader. Raw data are directly available in a spreadsheet file, which facilitates downstream calculations. In addition, the absorption of supernatant at another wavelength (e.g., at 340 nm, A_340_) is also recorded in order to correct for the potential light scattering that may arise from the contamination of the supernatant by Al_2_O_3_ particles, as illustrated in Figure 1A. Figure 1B shows the correspondence between the absorption at 260 nm minus the absorption at 340 nm (A_260_–A_340_) of the Al_2_O_3_ supernatant as a function of when the cAMP spiked into the standard assay medium. An excellent linearity is observed for cAMP, which varies from 25 to 400 μM. These values correspond to the conversion of 1.25% to 20% of the initial ATP substrate (2 mM), a range that is suitable for standard enzymatic assays. The time and concentration dependence of cAMP synthesis by the purified catalytic domain of CyaA (encoded by the first 384 residues of toxin, AC384 [28]) is shown Figure 1C. Given the high catalytic efficiency of AC384 (with k_cat_ in the range of 1500 to 2000 s^−1^) and its excellent time stability, this simple assay allows for an accurate determination (i.e., with robust absorption values) of its AC activity down to fmol amounts of purified AC enzyme.

### 2.2. CaM-Dependent Activation of AC and EF Activities

Figure 2 shows the CaM-dependence of the AC enzymatic activity of *B. pertussis* AC384 as well as that of a purified edema factor, EF, from the *B. cereus* (G9241 strain; this protein differs from the well-studied B. anthracis EF toxin by three amino acids, Pitard et al. manuscript in preparation). Both enzymes show high catalytic activities and high CaM-affinities in accordance with prior studies [25,29,30]. These data also indicate that the three amino acid replacements in *B. cereus* EF (I318T, V694A, and N789K, using amino acid numbers from *B anthracis* EF) have no impact on the catalytic efficiency or CaM affinity of the enzyme. This is consistent with the fact that they are located at the protein surface, away from the catalytic site and the CaM binding interface [31].

### 2.3. Enzymatic Assays in Complex Media

One advantage of this spectrophotometric assay is that it does not require any coupled enzyme to quantify the formation of the cAMP product. Therefore, it can be applied to measure AC activity in conditions that might affect the activity of many enzymes, like the presence of high salt or chaotropic agent concentrations. Figure 3A,B show the AC384 activity in the presence of an increasing concentration of NaCl or urea, respectively. A gradual decrease of activity is observed in both cases, although AC384 retains a significant catalytic efficiency (i.e., higher than 50% of its maximal k_cat_) even in the presence of up to 0.5 M NaCl or 0.2 M urea. We further examined the CaM-dependency of AC384 activity in the presence of 0.2 M NaCl or 0.2 M urea (Figure 3C) and found that although the maximal activity was reduced, the affinity for CaM was only marginally affected by these compounds. 

### 2.4. Characterization of a Detoxified CyaAE5-OVA Vaccine 

The non-radioactive AC assay can be useful for the characterization of CyaA-based vaccines. Indeed, earlier works have shown that detoxified recombinant CyaAs can be used as efficient antigen delivery vehicles. Various CyaA-based vaccines have been designed to trigger immune responses against infectious agents and/or cancer-specific antigens [32,33,34] (for a review [35]). The demonstration of the lack of AC activity (that is responsible for CyaA toxicity) of the detoxified CyaA vaccine is key in the process of GMP (Good Manufacturing Practice) preparation of protein lots for clinical evaluation. The spectrophotometric AC assay can be easily implemented for this, as illustrated in Figure 4, where a detoxified CyaA protein (CyaAE5, resulting from a specific mutation in catalytic site) carrying a model epitope (OVA, derived from ovalbumin), CyaAE5-OVA [36,37], was assayed at various concentrations. No traces of cAMP synthesis could be evidenced even with the highest concentration tested (200 nM CyaAE5-OVA, corresponding to 20 pmol per assay). Given that CyaAE5-OVA (as all CyaA proteins) was stored in a buffer containing ≈6.6 M urea to prevent the irreversible aggregation of the protein [5,38,39], we checked that the residual urea added to each assay (as indicated in Figure 4) did not affect the enzymatic detection by spiking sub-pmol amounts of active AC384. In all cases, high amounts of cAMP were easily detected. These experiments thus unambiguously demonstrate the total lack of AC activity of the purified CyaAE5-OVA preparation and indicate that this simple spectrophotometric AC assay could be easily implemented for control of GMP preparations of detoxified CyaA vaccines.

### 2.5. AC Assays in Crude Cell Extracts

To determine if the spectrophotometric assay could also be applied to unpurified or partially-purified AC preparations, or when the AC toxins would be diluted into complex cell extracts, we attempted to characterize CyaA binding to erythrocytes, which are frequently used as model target cells. Although these cells do not express the CyaA receptor (CD11b/CD18), CyaA can directly bind to the erythrocyte plasma membrane in a calcium- and temperature-dependent manner [40,41,42,43]. 

Purified CyaA was diluted in erythrocyte suspensions and incubated for 20 min at 30 °C in the presence of calcium (or in the presence of EDTA, or at 4 °C as controls). The cells were then extensively washed, resuspended in a small volume, and lysed with non-ionic detergent (e.g., Tween 20). The AC activities of the lysed samples were then assayed for various incubation times (0, 10, 20, and 30 min), and the absorption of the Al_2_O_3_ supernatants of all samples were recorded at different wavelengths (ie 260, 340, 405, and 595 nm). As anticipated, the cell lysates showed high absorption at 260, 340, or 405 nm (Figure 5). Yet, in control erythrocytes not incubated with CyaA, the absorption values remained constant over all the incubation periods (Figure 5A). In lysates of cells incubated with CyaA at 30 °C and in the presence of calcium (Figure 5B), the absorption at 260 nm (A_260_) increased linearly with time while the absorption at other wavelengths remained constant. By independently monitoring the cAMP content in the Al_2_O_3_ supernatants with an ELISA assay, we confirmed that the A_260_ increase precisely corresponded to the synthesis of cAMP due to the CyaA enzyme bound to cells (Figure 5B). The binding of CyaA to erythrocytes at 4 °C in the presence of calcium or at 30 °C in the absence of calcium (Figure 5C) was reduced in accordance with prior studies [41,42,43,44,45]. Hence, this spectrophotometric assay may be adapted to monitor AC toxin activity in complex cell extracts despite high absorption at useful wavelengths. 

## 3. Discussion

We show here that the direct spectrophotometric detection of cAMP after chromatographic separation on aluminum oxide can be used to provide a robust in vitro assay for bacterial AC toxin activities. This simple assay allows for an accurate determination (i.e., with robust absorption values) of the AC activity of *B. pertussis* CyaA. Thanks to the high catalytic efficiency (k_cat_ > 1000 s^−1^) and excellent time stability of this enzyme, we could detect down to the fmol amounts of purified AC. The assay is cost-effective and straightforward for routine monitoring of AC activity in safe conditions (i.e., without requiring manipulation of radioactivity). It can be easily set up in any laboratory without requiring expensive equipment or facilities (e.g., for manipulating radioactivity). It is perfectly appropriate for most standard assays of the bacterial nucleotidyl cyclase toxins that exhibit a high catalytic activity, such as the *B. pertussis* CyaA and *B. anthracis* EF—as shown here—and could be adapted as well for *P. aeruginosa* ExoY and other Gram- ExoY-like enzymes, which have turnover numbers in the range of 50 to 500 s^−1^ (depending on enzymes and/or substrates) [10,11,46,47]. Moreover, this spectrophotometric assay can also be applied to quantify the AC activity in complex mixtures, such as cell extracts, by shifting to a kinetic mode where the cAMP synthesis is measured via the specific increase of absorption at 260 nm (A_260_) as a function of the incubation time. Of course, proper controls should be performed to reliably establish that the A_260_ increase corresponds only to cAMP synthesis. It might also be adapted to characterize other less active nucleotidyl cyclases found in numerous bacterial or eukaryotic species—again provided that careful controls are applied.

Table 1 presents a comparison of the diverse assays that have been used to characterize bacterial AC toxins with their main advantages and drawbacks. The spectrometric assay described here lies in between the more sensitive and specific, but rather cumbersome, techniques that rely on cNMP detection (e.g., via radioactivity, ELISA, or LC/MS), and more high-throughput methods that might be less sensitive and more susceptible to interference by contaminating NTP- and/or pyrophosphate-metabolizing enzymes. Given its robustness and simplicity to set up, this in vitro spectrometric assay may thus become a method of choice for the routine characterization of the enzymatic activities of purified or partially-purified bacterial cyclase toxins. 

## 4. Materials and Methods

### 4.1. Materials

Aluminum oxide 90 active neutral (activity stage I) for column chromatography was obtained from Merck (ref # 101077, 70–230 mesh ASTM, Al_2_O_3_). UV-transparent 96-well microplates (Nunc™ UV 96 well) were purchased from Thermo Scientific (Waltham, MA, USA). ATP, cAMP, Bovine Serum Albumin, and Tween 20 were from Sigma-Aldrich (Saint-Louis, MO, USA). All recombinant proteins were expressed in *E. coli* and purified as described previously: CyaA protein [39,42], CyaAE5-OVA [36,37], AC384 (catalytic domain of CyaA, corresponding to residues 1 to 384), and CaM [28]. The EF enzyme tested here was from the *Bacillus cereus* G9241 strain (kind gift from P. L. Goossens, Pitard et al. manuscript in preparation) and comprises the adenylyl cyclase domain without the protective antigen-binding domain (i.e., residues 291 to 798 of full-length EF). This protein differs from the wild-type EF toxin from *B anthracis* by three amino acid changes: I318T, V694A, N789K (aa numbering from *B anthracis* EF shown in 1XFV.pdb, [48]). It was expressed in *E. coli* and purified as described by Drum et al. [31]. 

### 4.2. Enzymatic AC Assay

The enzymatic reactions were carried out in 1.5 mL Eppendorf tubes by sequentially adding: 50 μL of AC×2 buffer (Adenylate Cyclase assay buffer 2-times concentrated: 100 mM Tris-HCl, pH 8.0, 15 mM MgCl_2_, 0.2 mM CaCl_2_, 1 mg/mL bovine serum albumin (BSA), 10 μL of CaM (calmodulin) diluted in buffer D (dilution buffer: 10 mM Tris-HCl, pH 8.0, 0.1% Tween 20, or other non-ionic detergent such NP40 or Triton ×100) to appropriate concentrations (final concentration of 2 μM for standard assays), 10 to 30 μL of enzyme samples to be tested, diluted in buffer D to appropriate concentrations. Buffer D was added to a complete volume of 90 μL. The tubes were equilibrated at 30 °C for 3–5 min in a Thermomixer (Eppendorf, Montesson, France), and the enzymatic reactions were initiated by adding 10 μL of 20 mM ATP (stock solution in H_2_O, adjusted to pH ≈ 7.5). The tubes were incubated at 30 °C for appropriate incubation times (usually 5–20 min, but it could be extended to several hrs when needed, see Figure 1). A blank assay containing no enzyme (or no CaM) was carried out in parallel. The enzymatic reactions were stopped by adding 0.3 g (conveniently measured with a 0.2 mL Eppendorf tube) of dried aluminum oxide 90 active powder (the reaction was stopped immediately as the ATP binds to alumina). Alternatively, the enzymatic reaction could be stopped by adding 100 μL of 50 mM HEPES pH 7.5 or 50 mM EDTA (stop solution) to chelate the Mg^2+^ essential co-factor. Then, 0.9 mL (or 0.8 mL if 100 μL of stop solution have been added) of 20 mM Hepes-Na, pH 7.5 or 0.15 M NaCl (Chromatography buffer) were added to elute cAMP from the alumina powder. The tubes were mixed by inversion a few times over the course of 3–5 min and then centrifuged for 5 min at 12–14,000 rpm (room temperature) to pellet the alumina powder. The supernatants were carefully collected and the absorption at 260 and 340 nm were recorded. The cAMP concentration was then determined using an absorption coefficient at 260 nm of 15.4 mM^−1^ after subtracting the background absorption of the blank tube. The measurement of the absorption at 340 nm (A_340_) is very convenient for monitoring the potential light diffusion that may happen due to the presence of traces contaminating alumina powder in the supernatant solution. In our routine procedure, the OD_260_–OD_340_ was used to measure the cAMP content in each fraction.

A convenient format to carry out multiple assays is to transfer 0.3 mL aliquots of alumina supernatants into wells of a UV-transparent acrylic microplate, which allows for easy sequential measurements of the absorption at 260 and 340 nm (as well as other wavelengths see below) with direct data being recorded into an excel sheet. This greatly facilitates the downstream calculations. 

### 4.3. Assay of CyaA Binding to Erythrocytes

CyaA toxin binding to sheep erythrocytes was essentially assayed as previously described [42,43]. Sheep erythrocytes (from Charles River Laboratories, Wilmington, MA, USA) were washed and resuspended (≈5% of dry pellet) in HBS buffer (20 mM Hepes-Na, pH 7.5, 150 mM NaCl) supplemented with 5 mM glucose. The purified CyaA toxin (stored at 1 mg/mL in 8 M urea, 20 mM Hepes-Na) was directly diluted to 11.2 nM (2 μg/mL final concentration) into 1 mL of erythrocyte suspension supplemented with either 2 mM CaCl_2_ or 2 mM EDTA. A 20 μL aliquot was removed to determine the total adenylate cyclase activity added to each sample. The mixtures were then incubated at 30 °C, or 4 °C, for 30 min. The cell suspensions were chilled on ice, centrifuged at 4 °C, and cell pellets were washed two times with 1 mL cold HBS buffer (transferring the resuspended cells into new tubes to eliminate potential non-specific absorption of CyaA to the tube wall). Finally, the pelleted erythrocytes were resuspended in 100 μL of 10 mM Tris-HCl, pH 8.0, containing 0.1% Triton X−100 in order to lyse the cells. A total of 20 μL of lysates were tested in standard AC assays as described above and incubated for 0, 10, 20, or 30 min. The supernatants of aluminum oxide were then collected in a microplate and the absorption at 260, 340, 405, and 595 nm (the build-in filters available in our microplate reader–other wavelengths could be used as well in addition to the 260 nm) were successively recorded. Independent quantification of cAMP was performed on Al_2_O_3_ supernatants with an ELISA assay using a cAMP-biotinylated-BSA conjugate coated on ELISA plates and detected with a specific rabbit anti-cAMP antiserum [49].

## Figures and Tables

**Figure 1 toxins-14-00691-f001:**
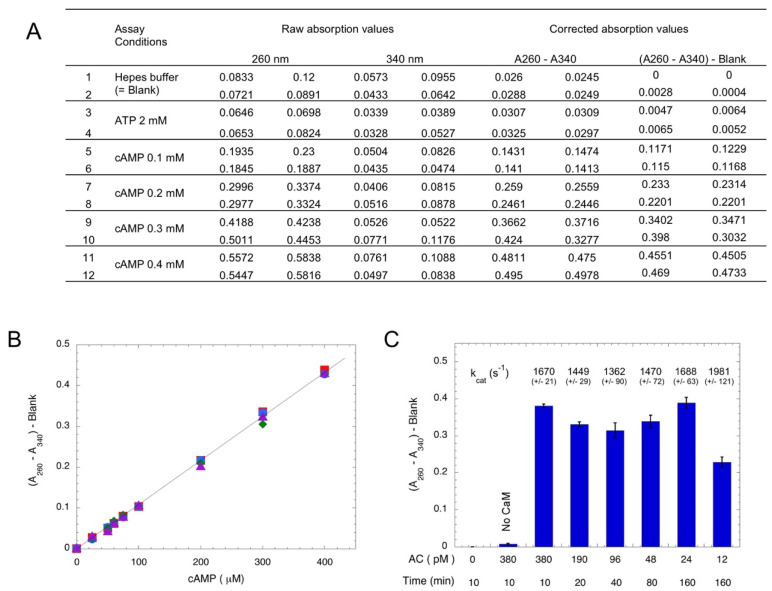
**Photometric detection of AC**. Panel (**A**): Spectrophotometric measurements of cAMP. Each line corresponds to an independent assay with the indicated nucleotide addition. Two 0.3 mL samples taken from the Al_2_O_3_ supernatant for each assay were transferred to a microplate and the absorption at 260 and 340 nm were recorded. The absorption at 340 nm provides a good estimate of the light scattering that arises from contaminating Al_2_O_3_ particles that are inadvertently aspirated in the 0.3 mL supernatant samples. Subtracting the absorption at 340 nm from the absorption at 260 nm yields a very reliable quantification of the cAMP content. Panel (**B**): Correspondence between the absorption at 260 nm minus absorption at 340 nm (A_260_–A_340_) of the Al_2_O_3_ supernatant as a function of cAMP (4 independent data points for each concentration) when it is added to the standard assay medium containing 2 mM ATP. Subtraction of the absorption at 340 nm allows for the correction of the potential light scattering arising from residual Al_2_O_3_ particles in the supernatant (see Material and methods for detail). Panel (**C**): Time and concentration dependence of *B. pertussis* AC activity. Reactions were carried out for the indicated times with the indicated final concentrations of AC384 in the presence of 2 mM ATP and 1 μM CaM (except in bar labeled “No CaM”). The calculated k_cat_ is indicated above the bar (mean of 4 independent experiments).

**Figure 2 toxins-14-00691-f002:**
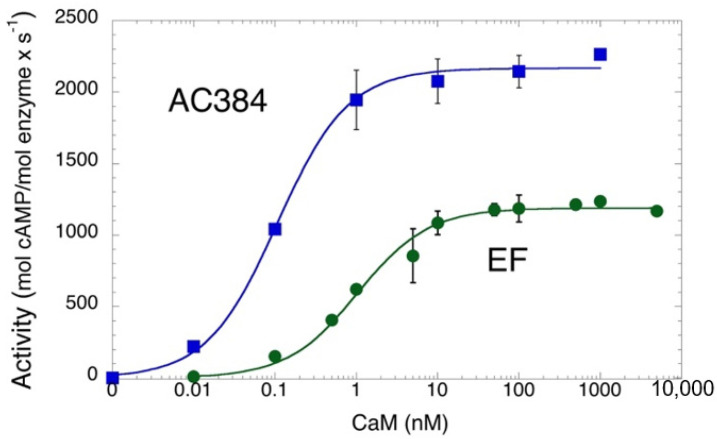
CaM-dependent AC activity of *B. pertussis* AC384 (blue) and *B. cereus* EF enzymes (green). AC activities (expressed in mol of cAMP produced per mol of enzyme per sec) at the indicated CaM concentrations were fitted to equation: A = (A_Max_ × [CaM])/(K_D_ + [CaM]). Maximal activities (A_Max)_ of 2165 ± 30 and 1183 ± 22 s^−1^ were calculated for AC384 and EF, respectively, while the CaM concentration at half-maximal activation (≈K_D_) were 0.11 ± 0.01 and 1.04 ± 0.13 nM for AC384 and EF, respectively.

**Figure 3 toxins-14-00691-f003:**
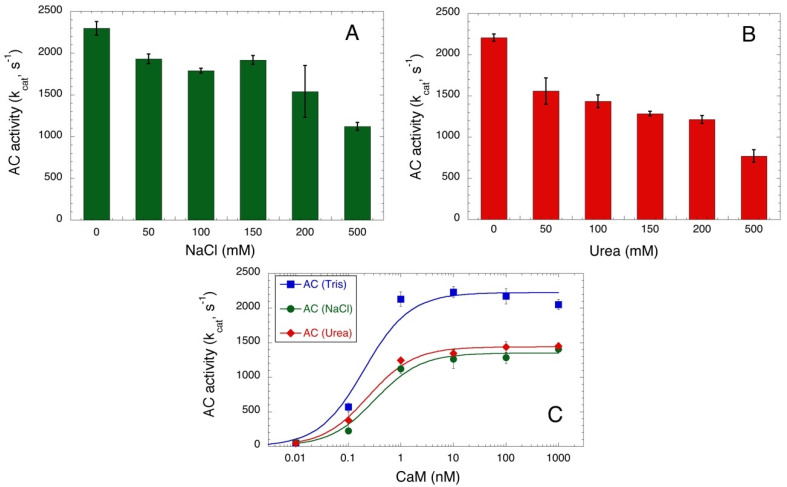
Salt and urea concentration dependence of AC384 activity. NaCl (Panel (**A**)) and urea (Panel (**B**)) concentration dependence of *B. pertussis* AC activity. Reactions were carried out for 10 min with 0.32 nM AC384 in the presence of 2 mM ATP and 1 μM CaM (mean and STD of 4 independent experiments). Control experiments (not shown) showed that ATP and cAMP binding to the Al_2_O_3_ were not modified by the presence of NaCl or urea at the tested concentrations. Panel (**C**): *B. pertussis* AC384 activities were measured at the indicated CaM concentrations in standard conditions (i.e., in Tris buffer, blue square) or in the presence of 0.2 M NaCl (green circle) or 0.2 M urea (red diamond). Data were fitted as in Figure 2. Maximal activities (in mol of cAMP per mol of AC384 per sec) were 2225 ± 95, 1351 ± 50, and 1440 ± 32 s^−1^ for assays in Tris buffer, in 0.2 M NaCl, and 0.2 M urea, respectively, while the CaM concentrations at half-maximal activation (≈K_D_) were 0.20 ± 0.06, 0.32 ± 0.08, and 0.24 ± 0.04 nM, respectively.

**Figure 4 toxins-14-00691-f004:**
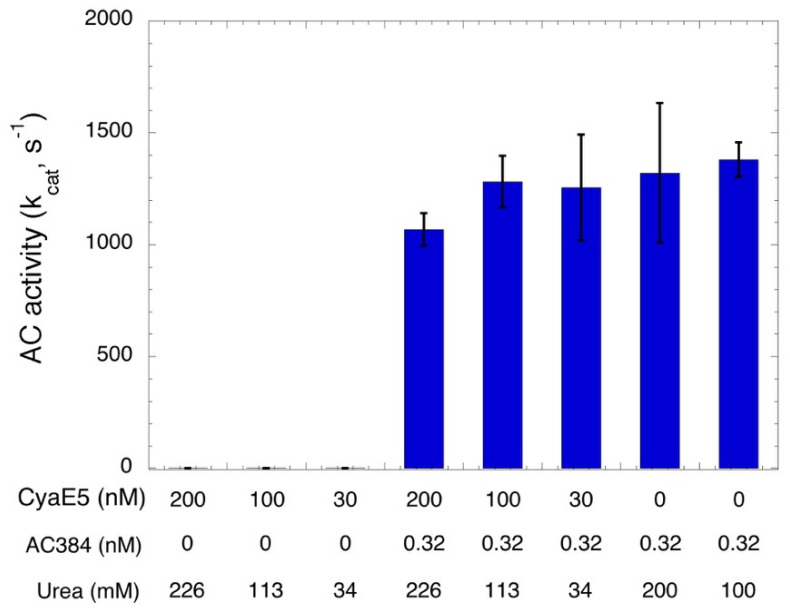
Characterization of a detoxified CyaAE5-OVA vaccine. AC activity of the purified CyaAE5-OVA protein (stored at 5.8 μM in 6.6 M urea, 20 mM Hepes-Na) at the indicated concentrations was monitored at 30 °C in the presence of 2 mM ATP and 1 μM CaM for 10 min (bars 1 to 6). As a control, in bars 4–6, 0.32 nM AC384 were spiked in the reaction mixtures in addition to the CyaAE5-OVA protein, in order to check that the residual urea concentration in each assay (indicated on the bottom line) did not affect the enzymatic activity. Results are the mean and STD of four independent experiments.

**Figure 5 toxins-14-00691-f005:**
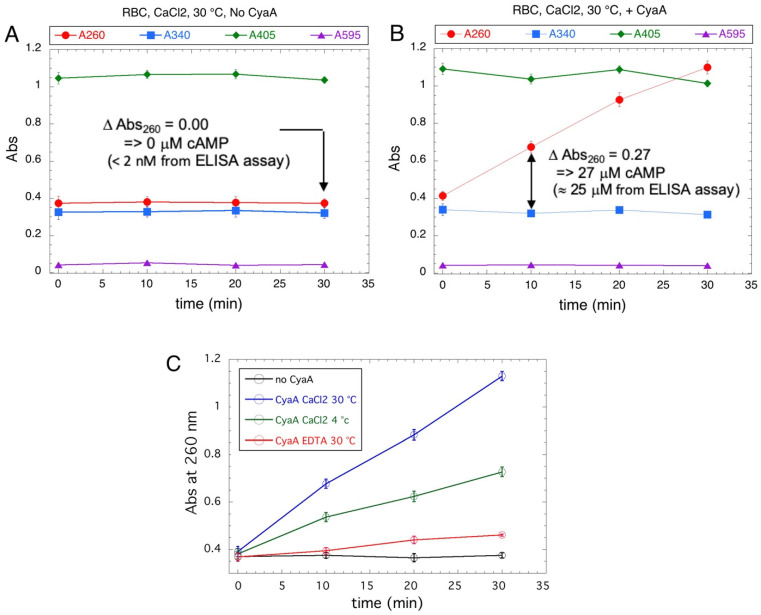
Assay of CyaA binding to erythrocyte. Panels A and B: Erythrocytes (RBC) were incubated at 30 °C with 2 mM CaCl_2_, for 30 min without CyaA (Panel (**A**)) or with 11.2 nM CyaA (panel (**B**)), and then extensively washed as described in the Methods section. After lysis, 20 μL of RBC lysates were tested in standard AC assays for the indicated times and the absorption of the Al_2_O_3_ supernatants were recorded at 260, 340, 405, and 595 nm. The cyclic AMP content in the Al_2_O_3_ supernatants was determined for sample A after 30 min of incubation and for sample B after 10 min of incubation by a specific ELISA assay and compared to the cAMP quantification deduced from the absorption at 260 nm (Abs_260_). Panel (**C**): Erythrocytes were incubated without (no CyaA, black) or with 11.2 nM CyaA at 4 or 30 °C with 2 mM CaCl_2_ or 2 mM EDTA for 30 min (as indicated). After extensive washing and cell lysis, 20 μL of RBC lysates were tested in standard AC assays as above for the indicated times. The relative fractions of “bound” CyaA activity, as compared to the total CyaA added to each cell suspension, correspond to about 2.4% for the incubation at 30 °C in the presence of CaCl_2_ (blue), about 1.1% for the incubation at 4 °C in the presence of CaCl_2_ (green), and less than 0.15% for the incubation at 30 °C in the presence of EDTA (red).

**Table 1 toxins-14-00691-t001:** Comparison of AC assay techniques.

Assay Principle	Characteristics	Drawbacks	References
* **Monitor production of cNMP product** *			
Radioactive: conversion of radioactive NTP into radioactive cNMP separated by Al_2_O_3_ chromatography and quantified in a scintillation counter	highly sensitive ^(a)^ (0.1 to 10 fmol) highly specific straightforward	Legal and environmental constraints of using radioactivity Expensive	[4,6,14]
Spectrometric: conversion of NTP into cNMP separated by Al_2_O_3_ chromatography and quantified by absorption at 260 nm	Sensitive (1 to 100 fmol of AC) straightforward inexpensive,	Possible application in complex media but requires proper controls	Present study
ELISA: Conversion of NTP into cNMP which is quantified by immunodetection	Highly Sensitive (0.01 to 10 fmol) Highly specific	Immunodetection (ELISA) assays are cumbersome and expensive Not high throughput	[15]
LC/MS: Conversion of NTP into cNMP which is quantified by liquid chromatography and Mass spectrometry	Highly sensitive (0.01 to 10 fmol) Highly specific Application to all cNMP	Special equipment (LC/MS) required Not high throughput	[16,17]
* **Monitor production of PPi product** * ** ^(b)^ **			
Colorimetric: detection of PPi product after hydrolysis into 2 × Pi quantified by colorimetry	Sensitive (30 to 100 fmol) Mid to high throughput	Not applicable in complex media or cell extracts ^(b)^ Requirement of coupled enzyme (PPiase)	[18,19]
Fluorometric: detection of PPi product after hydrolysis into 2 × Pi and coupling to production of fluorescent resorufin	Sensitive (1 to 100 fmol) Mid to high throughput Kinetic mode	Not applicable in complex media Requirement of coupled enzymes (PPiase, PNP, HRP)	[20]
Conductimetry: detection of PPi product after hydrolysis into 2 × Pi quantified by conductimetry	Low sensitivity (0.1 to 10 pmol) Kinetic mode	Not applicable in complex media Requirement of coupled enzyme (PPiase)	[21]
* **Monitor disappearance of ATP substrate** * ** ^(c)^ **			
Fluorescent: monitoring of ATP decrease with fluorescent probe Tb(III)–norfloxacin	Sensitive (10 to 100 fmol) Kinetic mode Mid to high throughput inexpensive, straightforward	Applicability in complex media? Possible interference with ATP consumption or production by other enzymes.	[22]
Luminescence: monitoring of ATP decrease with luciferase ^(d)^	Sensitive (10 to 100 fmol) Mid to high throughput	Applicability in complex media? Requirement of coupled enzyme (luciferase)	[23]
* **Reverse reaction: conversion of cAMP and PPi into ATP** *			
Spectrophotometric: ATP produced from cAMP and PPi-coupled by hexokinase and glucose-6-phosphate dehydrogenase-to the formation of NADPH measured by absorption at 334 nm	Low sensitivity (0.1 to 1 pmol) Kinetic mode Mid to high throughput	Interference with ATP production or consumption by other enzymes. Applicability in complex media? Requirement of coupled enzymes (hexokinase; glucose-6-phosphate dehydrogenase)	[24,25]

(a) Sensitivity: the values correspond to the lowest levels of CyaA (AC) or EF enzymes robustly detected in standard assays; (b) assays monitoring production of PPi product are usually not applicable in complex media or extracts due to presence of either Pi or Pi-generating reactions (any NTP hydrolyzing enzymes); (c) assays monitoring the consumption of ATP substrate will require that at least 20% of initial ATP is converted into cAMP in order to obtain reliable measurements. Moreover, these assays are usually not applicable in complex media or extracts due to the potential presence of ATP hydrolyzing or producing enzymes; and (d) in the luminescent assay described by Israeli et al. [23], there is an apparent discrepancy between the consumption of ATP (monitored by decrease of luminescence) and the appearance of cAMP which remains unclarified.

## Data Availability

Data is contained within the article.

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
