# Peer review of "A Robust and Sensitive Spectrophotometric Assay for the Enzymatic Activity of Bacterial Adenylate Cyclase Toxins"

_toxins, 2022, doi:10.3390/toxins14100691_

Round 1

Reviewer 1 Report

Adenylate cyclase toxins are key virulence factors of many bacterial pathogens. Once internalized into the host cell, they produce cAMP very effectively thereby subverting host cell signaling pathways and ultimately cell behaviour. Adenylate cyclase toxins have appeared as vaccine candidates, and some of them are being engineered as carrier plaforms for amino acid-based antigens to train host immune responses.  

This manuscript describes a new method to determine the enzymatic cAMP producing  activity of the adenylate cyclase toxins. 

Although a plethora of different assays are available for this same purpose, this new method has relatively good sensitivity, it is easy to perform basically in any laboratory, and, importantly, does not involve handling of radioactive material.  

Overall, the manuscript describes a solid method development study that is well described. I have only few comments with minor revision requests. 

Abstract and Introduction

Both of these paragraphs need to be revised to contain text describing the actual end use of this or the alternative methods to determine the enzymatic cAMP producing  activity of adenylate cyclase toxins. Why is it important to measure this activity? What kind of applications require this? 

Line 37

correct the spelling mistake in the bacterial name. 

Line 41

correct the wording, “implies to determine” is not english  

Line 79 and elsewhere in the manuscript

exchange the “rpm” with the corresponding g-value in centrifugation. 

Figure 1. 

Provide also A340 and A260 raw values. New sub-panel would probably do the job. It is important to display what kind of background does A340, i.e. aluminium particles, give. I want to see concrete justification for the A260-A340 substraction used throughout the study. 

Line 111

unify the way how the mutations are written in the text

Line 152

write what GMP means in brackets 

Reviewer 2 Report

This manuscript describes the advantages of the use of spectrophotometric assay for the enzymatic activity of bacterial AC toxins. The study design is appropriate and presentation of this manuscript is excellent. To make this article more complete, following points should be considered and revised.

1. line 70-72: Is there any reference of this sentence? (absorption at 260nm for AC) Otherwise, was this found by authors? Add evidence of the 260 nm absorption. May readers understand that absorption at 260 nm represents pure AC (or cNMP)? Is it common to AC from any bacterial species, or are there some differences ? This reviewer does not understand this point and thus it is better to add some explanations for readers in this regard.

2. Result section 2.1 is a long paragraph, so it is somewhat difficult to read. It is better to separate into some paragraphs. 

3. line 111:  What do I138>T, N789>K mean?

4. Table 1:  Is "Advantage" of upper heading appropriate? This reviewer thinks this may be "characteristics" or its related words. 

5. Discussion or Introduction: How do readers understand the significance of robust measurement method of bacterial AC? Is it used for fundamental basic research? Otherwise, will it be useful practically for treatment of bacterial infection, or any environmental investigation? Such overview of the present study may be preferable to be added.

Reviewer 3 Report

This manuscript is an interesting report describing a novel spectrometric assay for the adenylate cyclase toxin (AC) activity. The authors demonstrated that this assay after chromatographic separation on aluminum oxide accurately detected down to fmol amounts of B. pertussis CyaA and adapted to even crude cell extracts. This manuscript is fundamentally interesting and could potentially provide a new method for studying the characterization of AC and relative toxins. However, some points should be addressed.

Introduction

The authors described “for review” several times. These words are not necessary and should omit them.

Table 1

This table is busy and very difficult to read. Please make it more concise. For example, could this table be oriented landscape using one page?

Line 65

Why is the font of “Al2O3:” different, though the ones of ATP4- etc., are the same as other words?

Figure 5 and Line 315

CaCl2 should be CaCl2.
